

# PPE detector: a YOLO-based architecture to detect personal protective equipment (PPE) for construction sites

Md. Ferdous and Sk. Md. Masudul Ahsan

Department of Computer Science and Engineering, Khulna University of Engineering & Technology (KUET), Khulna, Bangladesh

## ABSTRACT

With numerous countermeasures, the number of deaths in the construction industry is still higher compared to other industries. Personal Protective Equipment (PPE) is constantly being improved to avoid these accidents, although workers intentionally or unintentionally forget to use such safety measures. It is challenging to manually run a safety check as the number of co-workers on a site can be large; however, it is a prime duty of the authority to provide maximum protection to the workers on the working site. From these motivations, we have created a computer vision (CV) based automatic PPE detection system that detects various types of PPE. This study also created a novel dataset named CHVG (four colored hardhats, vest, safety glass) containing eight different classes, including four colored hardhats, vest, safety glass, person body, and person head. The dataset contains 1,699 images and corresponding annotations of these eight classes. For the detection algorithm, this study has used the You Only Look Once (YOLO) family's anchor-free architecture, YOLOX, which yields better performance than the other object detection models within a satisfactory time interval. Moreover, this study found that the YOLOX-m model yields the highest mean average precision (mAP) than the other three versions of the YOLOX.

## INTRODUCTION

Due to urbanization, a significant number of workers are currently employed at construction sites to implement large projects in different parts of the world. Construction sites are always places of terrible danger. A small number of inspectors cannot take care of the health and safety of a large number of workers. According to the International Labour Organization (ILO), every year approximately 2.3 million people die due to work-related accidents or diseases (*Neale, 2013*). Every year at least 60 million terrible accidents happen in construction sites all over the world (*ILO, 2005*). It is estimated that an accident occurs every 10 minutes, which is a loud signal for the dangers of construction sites. One out of every six fatal calamities at the workplace happened at a construction site. In industrialized countries, as many as 25% to 40% of work-related deaths befall in construction sites (*ILO, 2005*). Most of the workers are under forty years of age. One study found that almost 25%

Corresponding authors
Md. Ferdous,
ferdous2007512@stud.kuet.ac.bd
Sk. Md. Masudul Ahsan,
smahsan@cse.kuet.ac.bd

of the workers had a mishap within a year of starting the work. In many countries, on an average 30% of the workers get into trouble from back pains or any other musculoskeletal disorders (*ILO, 2005*).

In construction sites there are more than 71% injuries compared to all other industry (*Waehrer et al., 2007*). However, workers can be protected from these types of terrible dangers by wearing personal protective equipment (PPE). Hardhats, safety glasses, gloves, safety vest, safety goggles, and so on are included as PPE. Workers can use hardhats to protect against minor head injuries. The chance of skull fracture, neck sprain, and concussion when falling from height can be diminished by wearing a hardhat (*Hume, Mills & Gilchrist, 1995*). It also reduces the possibility of severe brain injury. Hence, hardhat is an exigent part of the PPE in a construction site. Eye injuries are a very common phenomenon in the workplace, especially in construction sites. According to the National Institute for Occupational Safety and Health (NIOSH), approximately 2,000 workers in the U.S are suffering from a work-related eye injury. A study by the Bureau of Labor Statistics (BLS) shows that almost three out of five eye injured workers did not wear any protective shield at the time of the accident. A safety vest is another kind of PPE that helps a worker be more visible to other co-workers. Reflective strip lines of the vest may be helpful to extrapolate the location of the workers and reduce the chance of accidents in low lighting conditions and also in bad weather (*Wang et al., 2021b*). Hardhat colors may play a vital role to differentiate among the workers in different countries. In the United Kingdom (UK) black-colored hardhats are worn by the site supervisors, slinger/signallers wear an orange-colored hardhat, site managers wear a white-colored hardhat and the rest wear a blue-colored hardhat (*Wang et al., 2021b*). In the construction site, workers willingly or unwillingly forget to wear any element of PPE which may be dangerous for them or the whole construction site. Proper steps may diminish the risk of impending danger. The authority of the site should ensure that every worker wears PPE while they work in the construction site. However, manual checking would not be time and cost-effective. The background study found that correct detection of PPE is inevitable because misdetection or under-detection can cause a rigorous problem. From this motivation, the authors felt that precious detection of PPE can be helpful for workers' safety in an industrial manner. Moreover, additional PPE detection, *i.e.,* increasing the class number, increases the detection challenge in computer vision. That is why this study tries to recognize different types of PPE.

In this case, computer vision (CV) may be helpful. A system that can detect PPE in the construction site from the workers reduces the time and cost of the authority and improves the safety argument. For this purpose, this study created a new dataset named after CHVG that contains real-time construction site images. The primary objective of this study is to detect personal protective gear more accurately within a reasonable time interval. Moreover, misdetection or false detection would be harmful to both the workers and the authority of the construction site. The YOLO architecture is popular for the fast and accurate detection of objects from the image. An anchor-free manner architecture is published by Megvii (https://github.com/Megvii-BaseDetection/YOLOX) which is chosen

as the architecture to conduct this study. Satisfactory performance by the YOLOX are better than the previous study on the safety of the workers on the construction site.

A recent publication by *Wang et al. (2021b)* introduces YOLOv5 architecture for PPE detection into the construction site which detects six classes including four colored helmets, vests, and person. The authors of this study try to increase the reliability of CV and ensure more safety gear detection in a construction site by detecting eight classes. Therefore a new dataset is generated by extending (*Wang et al., 2021b*) proposed dataset. An anchor-free training architecture is introduced to detect PPE, person body, and person head in a construction site. Several photometric changes into the images are shown to create artificially rainy, hazy, and low-light conditioned images since the aforementioned situation would appear in a real construction site. YOLOX architecture yields better performance than the other state-of-the-art method.

## RELATED WORKS

In recent years several works have been done for the safety of the workers in construction sites based on CV and deep learning (DL). DL methods have a strong ability to self-learning from useful features. Region-based CNNs (R-CNNs) were used by *Fang et al. (2018)* to identify whether a worker wears a hardhat or not on the construction site. Their precision and recall rate was approximately 95.7% and 94.9% for the non-hardhat users (NHU). The authors did not consider hardhat detection rather they only detect the non-hardhat users. Histograms of oriented gradient (HOG) was used by *Zhu, Park & Elsafty (2015)* to extract head features from images. Feeding the extracted features into the support vector machine (SVM) the authors try to classify whether one worker wears a helmet or not. Single shot detector (SSD) based algorithm is proposed by *Wu et al. (2019)* to detect hardhat use. The authors found that the reverse progressive attention (RPA) network into the SSD can enhance the performance. Retinanet architecture is proposed by *Ferdous & Ahsan (2021)* to detect both hardhat and head of the workers in the construction site. The authors found the average precision (AP) for hardhat is 95.8% and the head is 93.8% for the publicly available dataset (*Xie, 2019*). SSD-Mobilenet was used by *Li et al. (2020)* to detect hardhat. The authors did not consider head detection as a safety issue. To identify protective helmets in the construction site modified SSD was presented by *Long, Cui & Zheng (2019)*. The AP of their model is 78.3% with 21.6 frame per seconds (FPS). A model was proposed by *Wang et al. (2020)* where the authors use MobileNet architecture as the backbone and residual block-based module for object prediction. The AP for hardhat is 87.4% and the head is 89.4% with a 62 FPS rate. *Mneymneh, Abbas & Khoury (2019)* tried to isolate a moving worker using the motion from videos then try to find any helmet on the top region. The color-based detection system was used by *Du, Shehata & Badawy (2011)*. They utilize color threshold to separate face, helmet, and other objects from an image. K-nearest neighbors (KNN) is used to capture moving objects from the videos then classification is done using CNN by *Wu & Zhao (2018)*. *Chen & Demachi (2020)* try to detect hardhats and full-face masks using YOLOv3 in Decommissioning of Fukushima Daiichi Nuclear Power Station.

The YOLOv3 architecture was used by *Delhi, Sankarlal & Thomas (2020)* to predict four classes such as NOT SAFE, SAFE, NoHardHat, and NoJacket. The authors trained their

model using 2,509 images that were collected from video recordings from the construction sites and internet-based collections. The average precision and recall rate is 96% on the test data. Authors try to make the CV more trustable in the real world by doing an alarm system by integrating it and also a reporting system with a certain time period. *Wang et al. (2021a)* used human identity recognition and helmet detection using YOLOv3 in a construction site. *Nath, Behzadan & Paal (2020)* presented a PPE detector using YOLOv3 algorithm. The authors proposed a dataset named Pictor-v3, which contains 1,500 images and corresponding annotations. They reported the highest performance is 72.3% mean average precision (mAP) with 11 FPS. The AP for vest and helmet is 84.96% and 79.81% respectively. *Zhang et al. (2021)* proposed an improved weighted bi-directional feature pyramid network (BiFPN) for hardhat wearing detection, they also try to detect the color of hardhat into the construction site. The authors showed 87.04% mAP yields by their proposed method for five classes. *Wang et al. (2021b)* present a dataset named CHV and use YOLO family architecture (YOLOv3, YOLOv4, and YOLOv5) to detect the PPE of the workers. The authors found that YOLOv5x outperforms the others method and the mAP is 86.55% for six classes. They proclaimed that 52 FPS for one single image is processed by the YOLOv5s model. This study is relevant and complementary to the aforementioned research. An attempt has been made to further enhancement of the above studies through this study. A new anchor-free training architecture is proposed for PPE detection into the construction site. This study also tries to explore the latest dataset for PPE detection both increasing data size and the number of classes.

## MATERIALS AND METHODS

### Dataset preparation (CHVG dataset)

Several sensor-based PPE detections have also been performed in the past years. In this study, a CV-based system is preferred since it is low costing, less complex, and easily usable at the field level than the sensor-based system. *Wang et al. (2021b)* present a dataset named CHV to detect the PPE of the workers. In the CHV dataset, there are six classes including four different helmet colors, vests, and persons. In a construction site, hardhat detection is important but also head detection is exigent. To make the CV more practical in an alarm system for non-hardhat users a major part is head detection. Every year more than 10,600 eye injuries disable the workers (*Thompson & Mollan, 2009*). Hence, we tried to detect person head and safety glass also as a part of PPE detection. To conduct this work, a new dataset was created named CHVG which consists of eight classes including four different colored hardhats (white, blue, red, and yellow), person head, vest, person body, and safety glass as an extension of the dataset named CHV. The name CHVG; CH for Color Hardhat, V for Vest, and G for Glass. Several images of the CHVG dataset are internet mined and most of the images were taken from *Xie (2019)* and *Wang et al. (2021b)*. Moreover, the hardhat is the major safety gear for workers to protect themselves from a minor accident. Hardhat color may play a different role in the construction site. The vest strip helps the authorities to observe workers who are located at a distance. In a construction area, it is a matter of tension when personnel are without a hardhat. Hence, person head detection

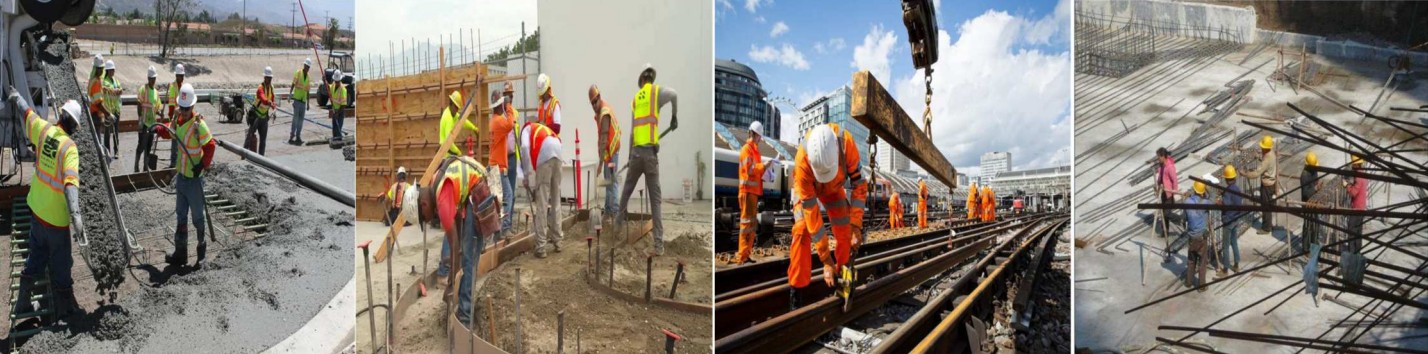

**Figure 1** Several images from CHVG dataset.

**Table 1** Dataset description.

| Class | No. of objects | Total No. of objects | No. of images |
|---|---|---|---|
| Person | 4,674 | | |
| Vest | 2,137 | | |
| Glass | 532 | | |
| Head | 730 | 11,604 | 1,699 |
| Red | 496 | | |
| Yellow | 1,200 | | |
| Blue | 543 | | |
| White | 1,292 | | |

would be a solution to remove this worry. Safety glass protect workers from harmful light rays, dust, air, gas, and other injuries. Several images from the CHVG dataset are shown in Fig. 1. The image size of the dataset is 640 × 640. The dataset can be found https://doi.org/10.6084/m9.figshare.19625166.v1.

The CHVG and CHV datasets are partially similar, but CHVG contains more 369 images than the CHV dataset. However, the CHVG dataset also contains extra two classes, such as safety glass and a head without a hard hat. In machine learning, more challenges for more classes. For this reason, we try to increase the class number. Besides, safety glass and head without hardhat detection on construction sites is essential, like other objects. After collecting the images, handcrafted annotations are done using the labelImg (*Tzutalin, 2018*) annotation tool. The CHVG dataset contains 1,699 images and corresponding annotations. The proposed dataset consists of 11,604 objects in total including 40.28% persons, 18.42% vests, 4.58% glass, 6.29% heads, 4.27% red, 10.34% yellow, 4.68%, and 11.13% white instances. Table 1 illustrates about the dataset. The CHVG dataset is divided into three subcategories *i.e.,* train, test, and validation set. Moreover, the test dataset for the model evaluation and the validation set is provided at the time of model training to see that whether the training is on the right path.

 

## The proposed framework

In this study, both training and testing are performed in an anchor-free manner object detection model of YOLO architecture named YOLOX. First of all, an image is fed into the trained model. Distinguishable features are extracted from the backbone of the architecture and fabricate a feature pyramid of the extracted features. Backbone is a feature extractor that represents the input image as a feature map. In this study, DarkNet53 (*Redmon & Farhadi, 2018*) is used as the backbone of the YOLOX architecture. Darknet-53 is a convolutional neural network (CNN) that serves as the backbone of the darknet YOLO architecture, which consists of 53 convolutional layers. More convolutional layers can learn more complex objects and works with higher accuracy.

Moreover, it can do more floating points operations than other models within less time (*Ge et al., 2021*). However, the dataset contains complex objects we need to do feature extraction with less time and better accuracy. For this reason, this study uses Darknet53 as the backbone of the experimented network. A feature pyramid is a pyramidal hierarchy of feature maps from low to high levels (*Ferdous & Ahsan, 2021*). It builds a pyramid of features, then transfers the feature pyramid to the head of the YOLOX architecture. Thereafter, the head network regresses the bounding boxes and classifies objects utilizing the features that come from the backbone. Figure 2 provides an overview of how the objects are detected from the images. The output can be any combination of the desired eight classes *i.e.,* person body, person head, vest, red, white, yellow, blue, and safety glass. The YOLOX architecture is shown in Fig. 3. In the object detection field, bounding box regression/object localization and object classification are done in parallel using approximately the same parameter (*Wu et al., 2020*; *Song, Liu & Wang, 2020*). A percussion rises up between these two tasks while trying to do object localization and object classification simultaneously. Basically, regression and classification sub-network uses nearly the same parameter to do their prediction task. Hence, two separate branches, one for classification and the other for localization *i.e.,* double-headed network was proposed in a double head R-CNN (*Chao et al., 2018*) to untangle the head of siblings. In a twin-headed network, object classification is done using a fully connected head network, and the object localization is done utilizing another convolutional head network (*Wu et al., 2020*). Due to having facilities in a double-headed network for object classification and localization many one-stage and two-stage object detection models follow dual-headed architecture (*Lin et al., 2017*; *Liu et al., 2018*; *Song, Liu & Wang, 2020*; *Chao et al., 2018*). According to Fig. 3, if we divide YOLO families architecture then it has three portions: backbone, head, and prediction.

Backbone (*e.g.,* Path Aggregation Network (PAN) (*Liu et al., 2018*) and the feature pyramid network (FPN) (*Kim et al., 2018*) continuously emits feature pyramids to the head. Using this feature, the head network classifies the objects and localizes the bounding boxes such that bounding boxes are aligned correct position of the objects. It is easy to switch YOLO architecture to an anchor-free mode. The head network needs to be changed slightly than previously published architecture. Reducing the predictions for each location from three to one and making them directly predict four values, *i.e.,* two offsets in terms of the left-top corner of the grid, and the height and width of the predicted box. Differently sized features are generated from the backbone of the architecture. Feature size can be defined as

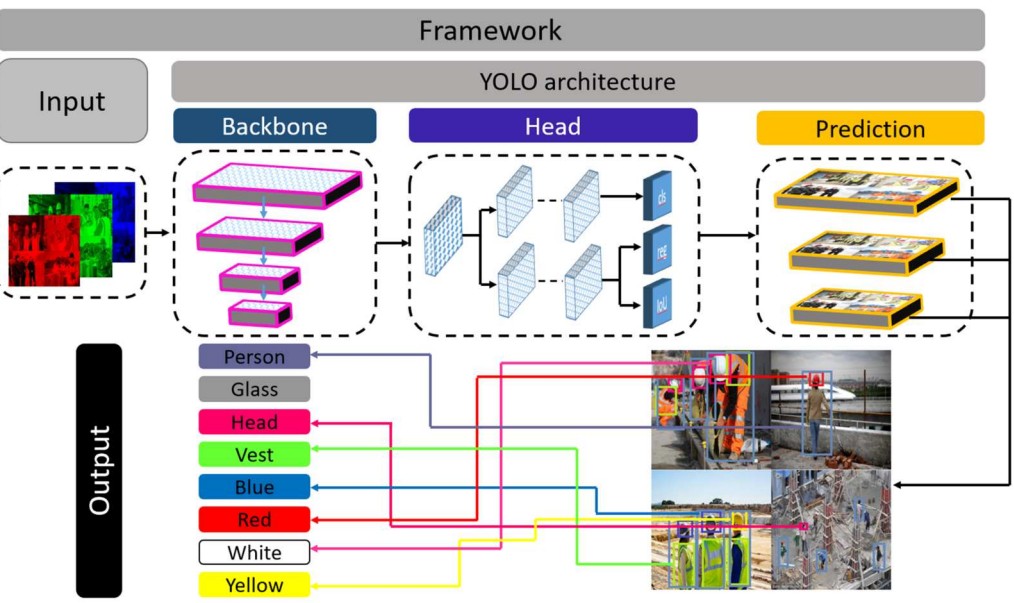

**Figure 2** Framework of the proposed system.

H × W × P where H and W are the height and width of the input image. The value of P can be 256, 512, or 1,024. Every feature map is sent to a 1 × 1 convolution layer. The output is added two parallel branches with two 3 × 3 convolution layers *i.e.,* one for classification and the other for regression. IoU branches are added to the regression branches for the IoU. The output of the classification, regression, and IoU branches are added to another 1 × 1 convolution layer for object classification, box regression, and IoU respectively. YOLOv3, YOLOv4 and YOLOv5 head network emits anchor boxes however, YOLOX (*Ge et al., 2021*) head network does not emit anchor boxes, hence it is said to be anchor-free manner architecture.

## TRAINING PROCESS

The CHVG dataset is divided into three different parts *i.e.,* training, testing, and validation. To check the robustness of the model we created three different combinations of the dataset by keeping the same amount of the images into the training, testing, and validation set. Figure 4 shows the object distribution of the dataset of a combination among the set. Figure 5 represents the distribution of the objects or instances according to eight classes of the dataset of a combination into the three different sets. For training the model this study ascertains the weight decay is 0.0005. Stochastic gradient descent (SGD) is used for function optimization, momentum is set to 0.9. The non-maximum suppression (NMS) threshold is adjusted to 0.65. NMS value is used to select an object's most appropriate bounding box. We can select the most appropriate bounding box at the testing time if we train the model using the most appropriate box. If we use a high NMS value, there is a possibility to detect more false negatives. Otherwise, a low NMS value yields more false

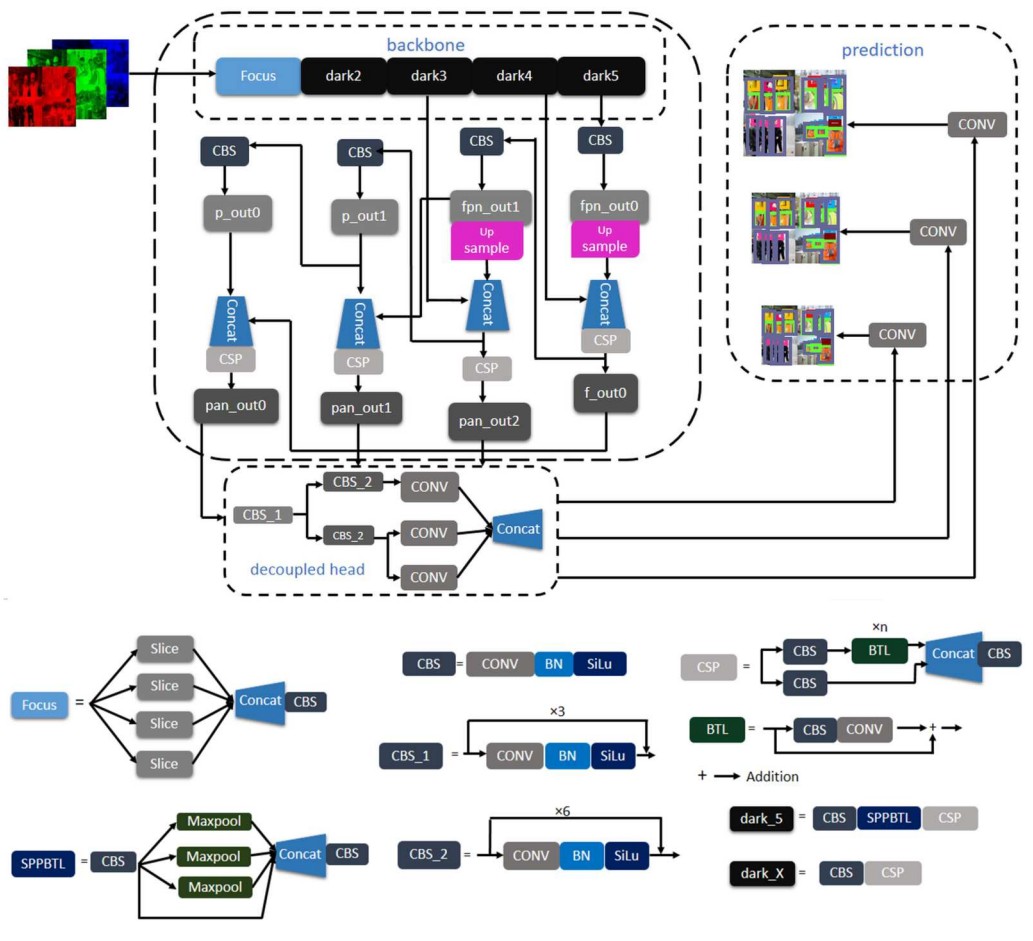

**Figure 3  Architecture of YOLOX.**

positive. For this reason, the authors use an NMS value of 0.65 at the time of training for validation process. The authors set the parameters from their interests. In addition, several parameters are set empirically. The first five epochs are warm-up epochs and the learning rate is 0.01. These warm-up epochs help the network to train gradually, making a basic sense of the dataset. The authors have experimented with different values however, the aforementioned setup performed well. Then the learning rate changes consecutively over time intervals according to the cosine learning rate schedule following the Eq. (1). A cosine learning rate schedule is used because it has a rapidly decreasing nature to a minimum learning rate value before being increased rapidly again. The resetting of the learning rate acts as a simulated restart of the learning process. Input image size is 640 × 640. The training process can be found in GitHub: https://github.com/Md-Ferdous/YOLOX. There are four versions of the YOLOX architecture: YOLOX-s (small), YOLOX-m (medium), YOLOX-l (large), and YOLOX-x (extra large). Batch size is agglutinate to 16, 12, 8, and 6 for the YOLOX-s, YOLOX-m, YOLOX-l, and YOLOX-x model respectively. All models are trained in the PyTorch environment. Table 2 represents the platform parameters. In this study, 200 epochs is performed to train the model. Table 3 represents the model description

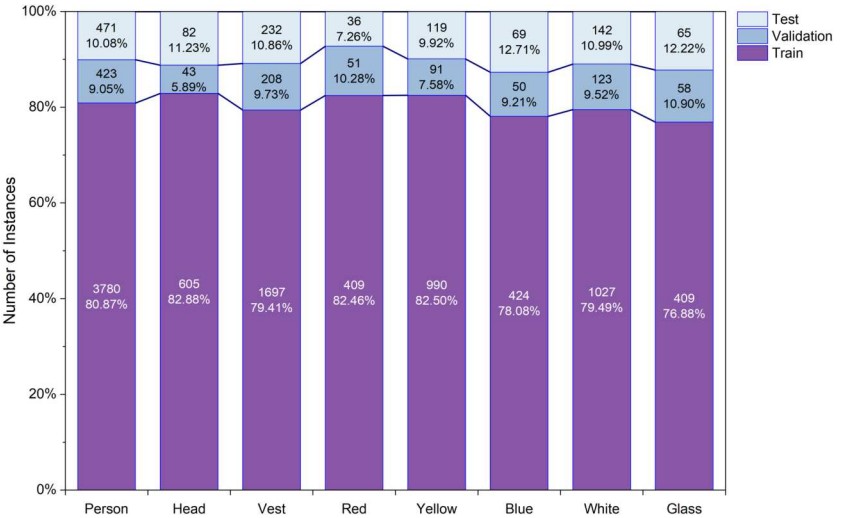

**Figure 4** Object distribution into training, testing and validation set.

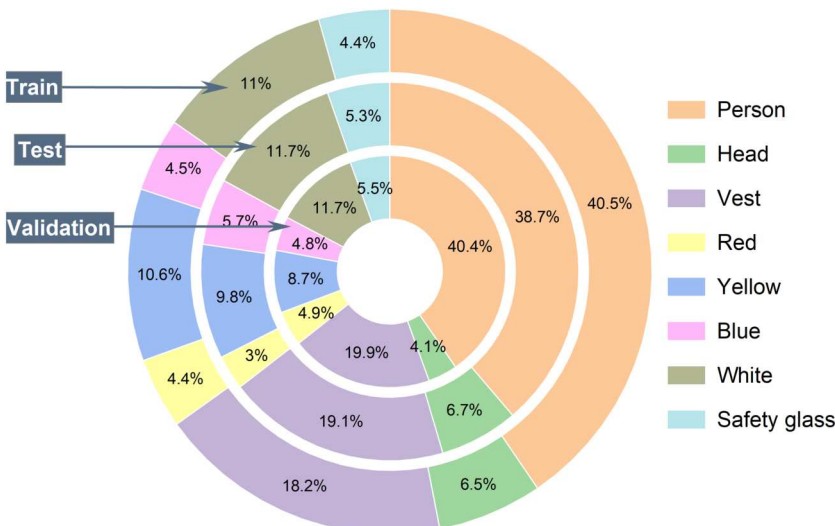

**Figure 5** The per class object distribution in each set.

of YOLOX architecture. Different versions of the YOLOX model depend on the model depth and width. Training parameter increases according to the model depth and width also. In this study, the training time of the YOLOX-m, YOLO-l, and YOLOX-x models increase approximately 2, 3, and 4 times subject to the YOLOX-s model training time.

$$lr = 0.5 \times \left(1.0 + \cos\left(pi \times \frac{\text{iteration}}{\text{total iteration}}\right)\right). \tag{1}$$

**Table 2  Platform parameters.**

| Platform | GPU | GPU size | CPU core | RAM |
|----------|-----|----------|----------|-----|
| Training | NVIDIA GeForce RTX 2080 Ti | 11 GB | 20 | 128 GB |
| Testing | NVIDIA Tesla K80 | 11 GB | 2 | 12 GB |

**Table 3  Model description.**

| Model | Depth | Width | Parameters (M) | Training time (hours) |
|-------|-------|-------|----------------|-----------------------|
| YOLOX-s | 0.33 | 0.50 | 9 | 4 |
| YOLOX-m | 0.67 | 0.75 | 25.3 | 8 |
| YOLOX-l | 1.0 | 1.0 | 54.2 | 11 |
| YOLOX-x | 1.33 | 1.25 | 99.1 | 15 |

## Data preprocessing

Several geometric changes of the images are performed as the mosaic augmentation, the random affine transformation is performed where rotation is accomplished on both axis using a value between ($-10$ deg to $+10$ deg). Both $X$ and $Y$-axis translation is performed within a value of (0.4, 0.6). Scaling is done on both axis using a value between (0.1, 2), the same amount of shear is performed on both axis. The quantity of shear is accomplished within a value of ($-2$ to $+2$). All aforesaid values are uniform random values between the range. Photometric changes are executed by changing the brightness, contrast, hue, and saturation. Moreover, RandomHorizontalFlip and RandomResizedCrop strategies were also performed as the data preprocessing.

## HAZE, RAIN AND LOW-LIGHT EFFECT

Different types of challenging conditions are revealed in construction sites that creates problem for a machine to recognize an object properly from the scene. In such situations, humans can make small mistakes, so keep in mind the conditions that may be created in real-time; this study tries to generate artificial images by doing several photometric changes into the image. These artificial images are not used at the model's training time. They are used only for the testing purpose. The HAZE, RAIN, and LOW-LIGHT images are created from the test dataset to see the model's robustness. The effected images are not included in our dataset. According to the background study, this study found a variety of challenging conditions may appear in real-time construction site e.x. low-light, rain, and haze effect are some of them.

Algorithm 1 shows how to add artificial haze to make a hazy image. First of all, we take a 2D array of the same size as the original image to create a hazy image. The real hazy image contains some noise, hence to make the artificially created image more realistic we add some noise to it. Noise amplitude controls the severity of the noise and noise offset controls the brightness of the noise. This study adds pixel by pixel values of noisy images and original images to make the hazy images. The same noise is added in all color channels so that it does not change the color property. Then adjust the pixel values of the hazy image using maximum pixel values of the hazy image and original image.

---

**Algorithm 1** HAZY-IMAGE

---

**Require:** image, noiseAmplitude, noiseOffset

    $noiseImage \leftarrow$ Generate 2D random array of same size of input image

    $noiseImage \leftarrow noiseAmplitude \times noiseImage + noiseOffset$

    $maxInput \leftarrow max(image)$

    $hazeImage \leftarrow image + noiseImage + 250$

    $maxHaze \leftarrow max(hazeImage)$

    $hazeImage \leftarrow hazeImage \times maxInput/maxHaze$

---

Algorithm 2 and 3 shows how to appear artificial rain to make rainy image. This study uses randomly value between $(-10, 10)$ to make the raindrops slant. The height and width of the raindrop are taken 2% and 0.18% subject to the height and width of the original image size respectively. The color of the raindrop is set to RGB (196, 211, 223). All values are taken from the author's perspective. Then the predefined number of raindrops is drawn on the image. To make the image a real rainwater image, this study applies an average kernel of size 4 on the image as a result, the image is a bit blurry. To make the image a little shady we reduce the brightness by 30% less than the real image because the rainy image is shady than the non-rainy image.

---

**Algorithm 2** GENERATE-RANDOM-LINES

---

**Require:** imageShape, slant, dropLength

**Ensure:** $drops \leftarrow \Phi, i = 0$

    $numDrops \leftarrow$ Total number of rain-drops

    $noiseImage \leftarrow$ Generate 2D random array of same size of the input image

    **while** $i < numDrops$ **do**

        **if** $slant < 0$ **then**

            $x \leftarrow random(slant, imageShape[1])$

        **else**

            $x \leftarrow random(slant, imageShape[1] - slant)$

        **end if**

        $y \leftarrow random(0, imageShape[0] - dropLength)$

        $drops \leftarrow append(x, y)$

    **end while**

---

This study creates a low-light image by controlling the brightness of the original image. Reducing 60% brightness subject to the original image to create a low-light conditioned image. Although light angles may differ in real-time, however, this study reduces the brightness of every pixel linearly. Figure 6 shows artificially created hazy, rainy and low-light image.

## EVALUATION METRICS

In object detection, we should take the high value of both precision (*Padilla et al., 2021*) and recall (*Padilla et al., 2021*). Unfortunately, sometimes this is a matter of concern to

---

---

**Algorithm 3** RAINY-IMAGE

**Require:** image

**Ensure:** $rainDrop = 0$

   $imageShape \leftarrow shape(image)$

   $slant \leftarrow random(-10, 10)$

   $dropLength \leftarrow int(imageShape[0] \times 0.02)$                              ▷ $2\%\,of\,height$

   $dropWidth \leftarrow int(imageShape[1] \times 0.0018)$                       ▷ $0.18\%\,of\,width$

   $rainDrops \leftarrow GENERATE - RANDOM - LINES(imageShape, slant, dropLength)$

   **while** $rainDrop \neq rainDrops$ **do**

      $rainyImage \leftarrow$ Draw rain drops on image

   **end while**

   $rainyImage \leftarrow$ Apply blur filter to rainyImage

   $rainyImage \leftarrow$ Reduce brightness of rainyImage

---

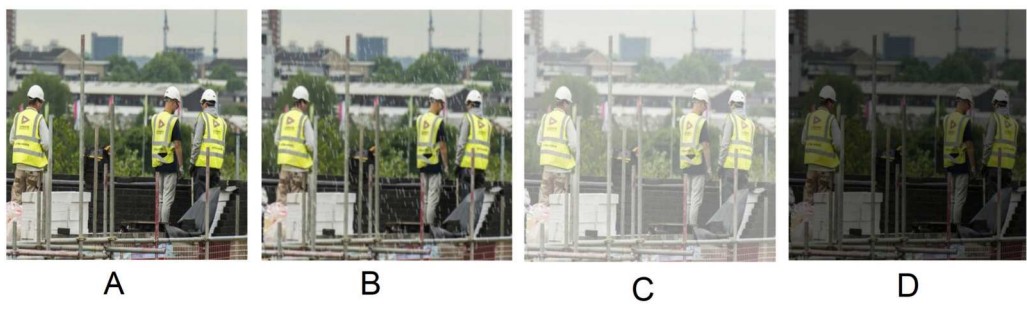

**Figure 6** Artificially created rainy (B), hazy (C) and low-light (D) images. (A) is the original image.

take the best value of both the metrics. In this regard, the precision-recall (PR) curve may help us to select the best value for evaluating a model. In the PR curve, precision is plotted on $Y$-axis, and recall is plotted on $X$-axis, therefore the optimum value for both the metrics can be acquired in the right up corner.

In object detection, NMS is calculated for selecting one entity from many overlapping entities e.x. one bounding box is to be selected from numerous overlapping bounding boxes. The intersection over union (IoU) calculation may help us to pick up the best bounding box from multiple boxes. Precision, recall and IoU can be calculated according to *Padilla et al. (2021)* and *Padilla, Netto & da Silva (2020)*. The IoU calculation has another effective use in the performance measurement of an object detection model. When we try to measure the performance of a model, we must look at the appropriate object and perfect bounding box alignment around the object. The IoU computation could assist us in choosing a better bounding box for indefectible alignment. The IoU calculation can be performed using the ground truth bounding box and the predicted bounding box. IoU calculation tells us how much the predicted bounding boxes are related to the ground truth bounding boxes *i.e.,* the percentage of overlap between two bounding boxes. The bigger the overlap area, the higher the IoU. TP, FP, and FN are calculated according to *Padilla et al. (2021)*. In this

study, the authors consider a detection as TP if the model predicts true class and the IoU calculation of bounding boxes is greater than 50%. Inversely, if the IoU is smaller than 50% and the detection emits the right class according to ground truth then it is considered as FP. FP detects a ground truth class however, the ground truth box and the predicted bounding box is not in the same position. FP yields an improper detection case. In the case of FN, the system won't be able to detect any class where ground truth boxes exist. We can calculate FN by subtracting TP from the total positive. For calculating the TP, FP and FN this study uses object confidence score is greater than or equal to 50%.

AP is another metric to evaluate an object detection model. This study tries to evaluate the performance of the model using the AP. AP is a single value that illustrates the average of all precision. AP is also a scheme to encapsulate the PR curve. Higher precision is a piece of clear evidence that a model is confident while it classifies instances among the detections. On the other hand, higher recall is an indicator of the power of a model, it tells us how many correct detections are performed among all the ground truths. Moreover, both precision and recall are major metrics of an object detection model. If a model has high recall yet low precision is an obvious referential that the model emits maximum positive example truly but it has many false positives *i.e.,* classify many negative examples as positive. On the contrary, low recall yet high precision is an indicator that the model appropriately classifies positive examples however, it may contain only a few positive examples. Hence, it is necessary to choose a threshold, as if both precision and recall will be maximized. The PR curve helps to select the threshold among the different threshold values. Using the precision and recall value, the PR curve can be plotted (*Padilla et al., 2021*). AP is the area under the curve (AUC) of the PR curve. AP and mAP can be calculated according to *Padilla, Netto & da Silva (2020)* and *Padilla et al. (2021)*.

## RESULTS

All models of the YOLOX architecture were trained and tested using three different combinations of the dataset to check the robustness of the model. Table 4 represents the AP and mAP of the eight different classes. The YOLOX-m model yields the highest mAP applying the second combination of the dataset. Moreover, averaging the mAP of the three different dataset combinations of every model, the YOLOX-m model performs better than the other two combinations of the dataset and generates the highest average of mAP is 89.84%.

Figure 7 represents the PR curve of the YOLOX architecture. From this curve, it is seen that the best performance by the model yields at the top right corner *i.e.,* whether the value of the top right corner of the curve is chosen then both the precision and recall will be maximized.

Figure 8 displays the TP, FP, and FN rate of every class of the YOLOX-m model. The highest TP rate is 95.94% for the person object and the lowest TP rate is 86.15% for the safety glass object. The highest FP rate is 7.38% for the white object and the lowest FP rate

**Table 4 Performance of YOLOX in the different dataset combinations.**

| Combinations | Person | Head | Vest | Red | Yellow | Blue | White | Glass | mAP | Avg. of the mAP |
|---|---|---|---|---|---|---|---|---|---|---|
| | | | | | YOLOX-s | | | | | |
| 1 | 93.19 | 80.21 | **93.55** | 89.63 | 86.25 | 90.01 | 91.12 | 79.95 | 87.99 | |
| 2 | 92.89 | 86.90 | 88.14 | 89.72 | 92.77 | 90.19 | 93.43 | 78.28 | 89.04 | 88.19 |
| 3 | 92.76 | 90.61 | 88.72 | 85.33 | 91.17 | 85.15 | 92.55 | 74.12 | 87.55 | |
| | | | | | YOLOX-m | | | | | |
| 1 | 94.54 | 90.42 | 93.41 | 90.65 | 89.14 | 84.69 | 93.71 | 78.93 | 89.44 | |
| 2 | 94.72 | 89.85 | 90.28 | 85.54 | 92.06 | 88.08 | 93.17 | **85.00** | **89.84** | **89.47** |
| 3 | 93.74 | 87.45 | 89.88 | **90.85** | 91.29 | 86.66 | 92.39 | 81.86 | 89.27 | |
| | | | | | YOLOX-l | | | | | |
| 1 | 93.53 | 90.84 | 90.07 | 89.60 | 89.14 | 87.35 | 94.52 | 77.85 | 89.11 | |
| 2 | 94.86 | **92.37** | 89.54 | 88.51 | 92.29 | 86.57 | 93.29 | 79.28 | 89.59 | 89.11 |
| 3 | 93.39 | 86.04 | 88.28 | 89.51 | 92.45 | 86.29 | 93.86 | 79.28 | 88.64 | |
| | | | | | YOLOX-x | | | | | |
| 1 | 92.82 | 90.16 | 89.47 | 85.49 | 89.00 | 88.91 | 96.18 | 84.25 | 89.53 | |
| 2 | **95.41** | 78.19 | 90.67 | 86.80 | **95.45** | **92.56** | **96.45** | 81.66 | 89.65 | 89.35 |
| 3 | 90.69 | 86.79 | 88.77 | 88.47 | 90.50 | 86.30 | 95.18 | 84.40 | 88.88 | |

**Notes.**
Bold values indicate the highest values of every column.

is 2.74% for the head objects. The highest FN rate is 13.85% for glass objects and the lowest FN rate is 4.06% for the person objects.

Figure 9 represents the inference time on both the GPU and CPU for a single image. From this figure, it is seen that increasing the model size increases the inference time on both the GPU and CPU. In the GPU, YOLOX-s takes almost 0.08s time to infer an image whereas in the CPU it needs 0.7s to accomplish inference of a single image which is 8.75 times slower than the GPU. GPU inference time may differ based on the architecture of the GPU. The reported time is generated in NVIDIA Tesla K80 GPU.

Figure 10 represents several qualitative results of different versions of the YOLOX architecture. According to Fig. 10, a misdetection appears by the YOLOX-l and YOLOX-x model. Although YOLOX-s and YOLOX-m model soothing this misdetection. A white hardhat with a headphone is detected correctly by the YOLOX-s and YOLOX-m model whereas YOLOX-l and YOLOX-x model does not detect this object *i.e.,* a false negative occurs.

Figure 11 shows the mAP variations subject to the different sizes of the test image set. The YOLOX-m both trained and tested with the 640 × 640 image size. To check the robustness of the model this study both increase and decrease the ratio of 20% and 35% size of the test image subject to the original image size of 640 × 640. Basically, reducing the size of the image, the smaller objects become smaller hence, there is a possibility of under-detection of an object by the model. Furthermore, sometimes false detection arises which affects the performance of the model. Therefore, the mAP reduces with the decreasing of the image size. On the contrary, when we increase the image size, the nearest objects move closer, consequently, they are undetected or false detection arises by the model. Therefore, the

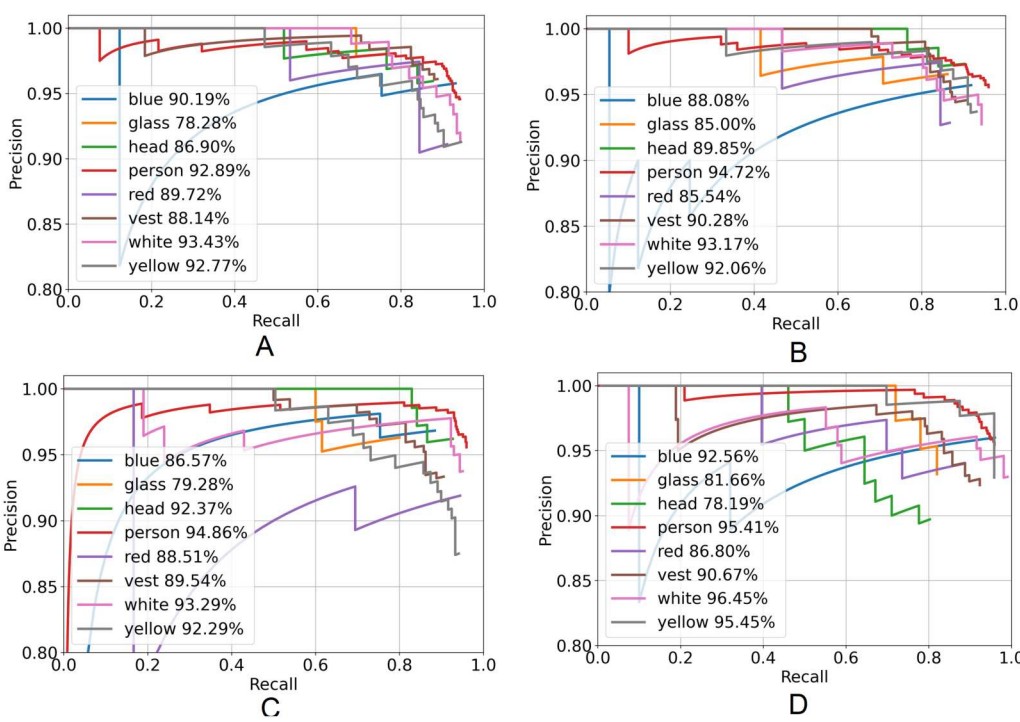

**Figure 7  Precision–recall curve of YOLOX architecture.** (A) YOLOX-s, (B) YOLOX-m, (C) YOLOX-l and (D) YOLOX-x.

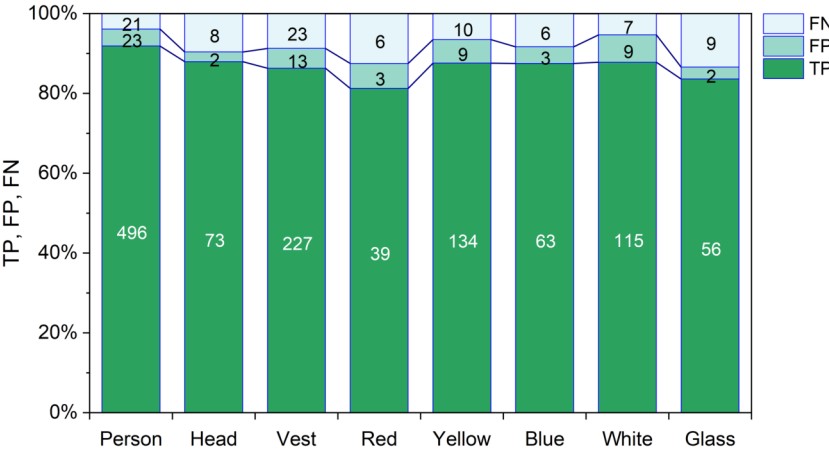

**Figure 8  TP, FP and FN of YOLOX-m.**

mAP reduces with increasing the image size. Moreover, in our case 35% size increasing subject to the original image size *i.e.,* increased image size is 864 × 864 yields 88.89% mAP. Whereas increasing the image size subject to original image size *i.e.,* increased image size is

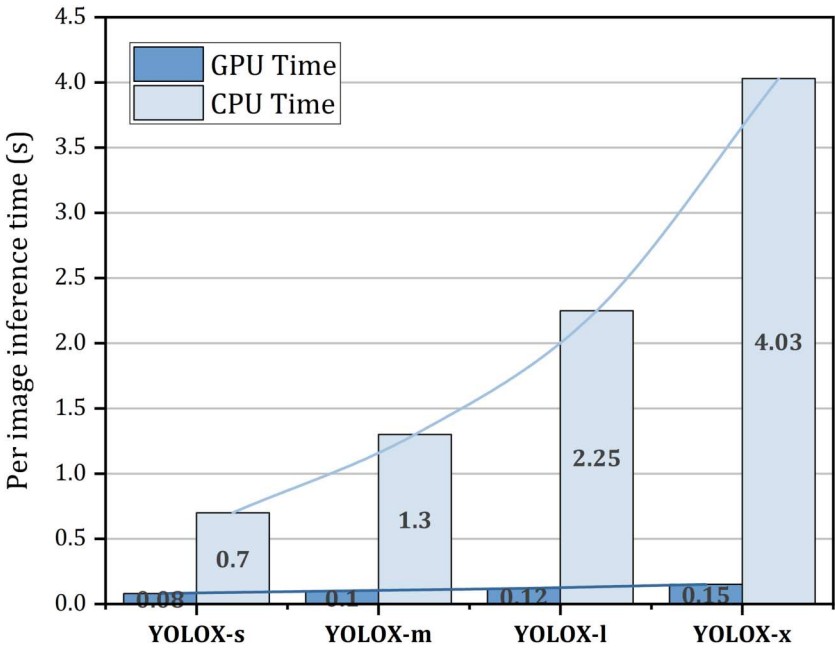

**Figure 9** Per image inference time on both GPU and CPU.

768 × 768 yields 88.55% mAP. Most of the images in this study are taken from real-time construction sites where the camera position is not closer to the scene. This may be due to the fact that the mAP did not decrease significantly, though the image size increases. When we decrease the image size of 20% and 35%, the mAP also decreases by 2.48% and 5.24% respectively. Image resize and corresponding annotations are done using roboflow (https://roboflow.com/).

Figure 12 represents several satisfactory results of the YOLOX-m architecture. In Fig. 12A objects are in dense condition *i.e.,* objects are occluded, even after they are properly detected. Objects are lined up one after another with an angle Figures 12G yet they are detected accurately. Figs. 12A, 12B and 12H look like pictures taken by a CCTV camera *i.e.,* the camera position and image scene are not closer; as a result, objects are small even though unerring detection yields by the model. Objects are natural working mode, for example kneeling down, bending the spine and different angles, even several objects on their knees in Figs. 12D, 12E, 12F and 12I still accurate detection is generated by the model.

Figure 13 represents several incorrect detections of the YOLOX-m architecture. In Fig. 13A a false detection appears, something is detected as a vest that is not actually a vest. In Fig. 13B, a vest object is not detected. In Fig. 13C, a yellow color part of a machine is detected as a yellow hardhat.

Table 5 shows the quantitative measures of the YOLOX-m on low-light, hazy, and rainy effect images. The mAP of the Low-light, Hazy, and Rainy effect images is 3.90%, 2.10%, and 15.20% less than the original image respectively. Here, it is seen that objects are not correctly detected while we test the model using the rainy effect images. This is because there is a difference between the actual rainwater image and the artificially created rainy

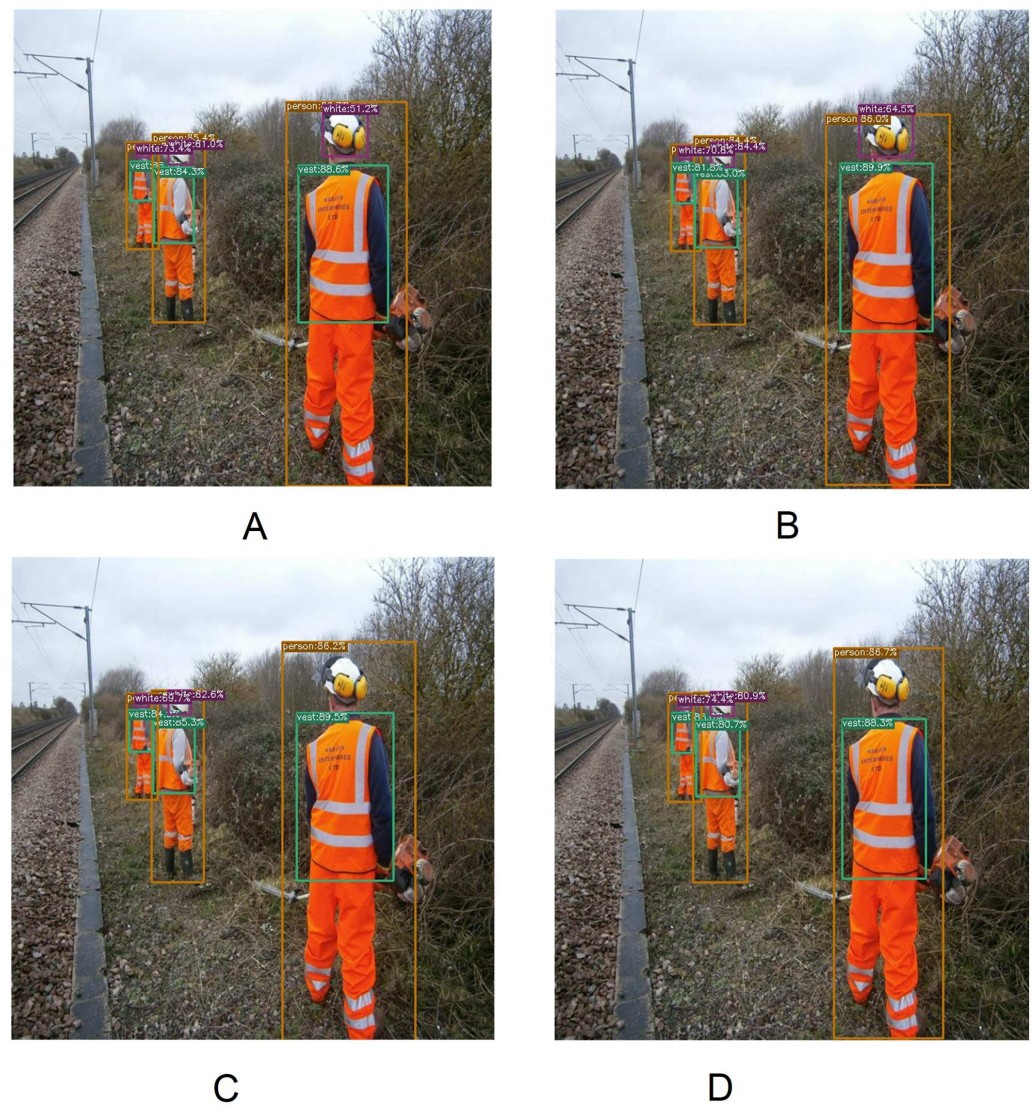

**Figure 10  Several qualitative measures of YOLOX.** (A) YOLOX-s, (B) YOLOX-m, (C) YOLOX-l and (D) YOLOX-x.

image. Moreover, if we had used these artificially created rainwater images at the time of model training, they might have performed even better at the time of testing. Figure 14 displays several quantitative measures of Low-light, Hazy, and Rainy effect images of the YOLOX-m model.

## Comparison with the state-of-the-art

In this section, a comparison is illustrated with the state-of-the-art method. According to the background study, this study found that *Wang et al. (2021b)* proposed method performs better than the other state-of-the-art methods. Therefore, this study shows a comparison with the *Wang et al. (2021b)* achievement. When we apply YOLOX-m to the

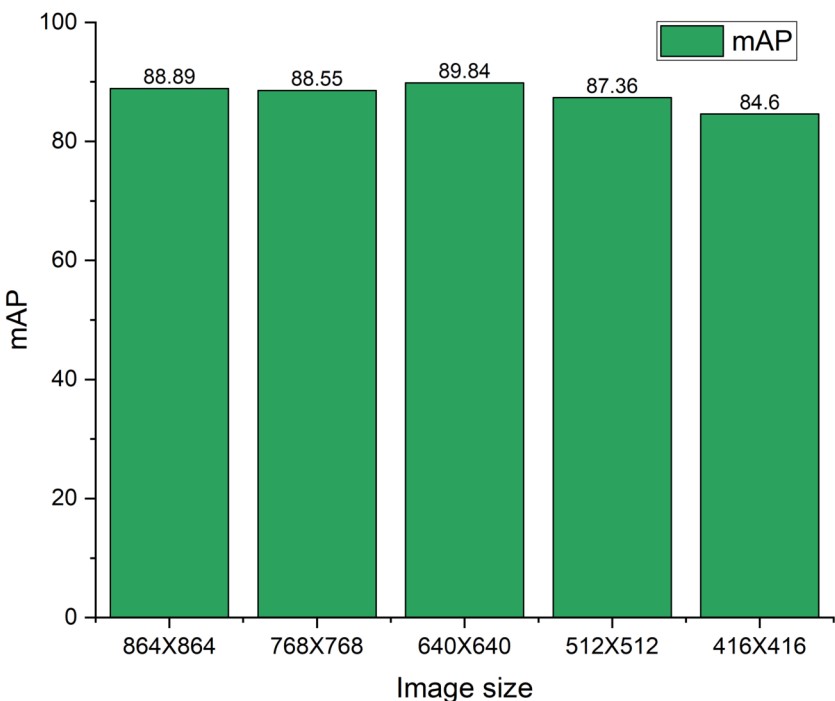

**Figure 11** mAP variations on different test image sizes.

*Wang et al. (2021b)* dataset named CHV, there were approximately 3.51% mAP increases than their proposed method named YOLOv5x. The analogy is shown in Table 6. In addition, YOLOv5x is both trained and tested with our CHVG dataset. YOLOv5x yields 86.24% mAP on the testing dataset, whereas this study's proposed method YOLOX-m delivers 89.84% mAP on the testing dataset which is approximately 3.60% improvement than the YOLOv5x method.

Table 7 shows a disparity of blurring face tests between YOLOv5x and YOLOX-m. Moreover, for the blurring face test, this study ignores the safety glass detection case. *Wang et al. (2021b)* reported 79.55% mAP yields using the YOLOv5x model on their CHV dataset. However, this study's proposed method YOLOX-m generates 88.78% mAP. More generally which is nearly 9.23% higher than YOLOv5x. This study blurs a face using the pixelization process on a certain face region.

## DISCUSSION

In this work, a new anchor-free manner architecture YOLOX from the YOLO family is used to detect PPE in a construction site. A new dataset is created by extending the CHV (*Wang et al., 2021b*) dataset where the authors try to detect six classes. This study increases the dataset size by increasing the corresponding images of construction sites and trying to detect eight classes. The YOLOX performs better than other state-of-the-art methods. This study found that YOLOX-m yields the highest mAP of 89.84% among the other three versions of the YOLOX. Per image inference time of YOLOX-m was approximately 0.99s.

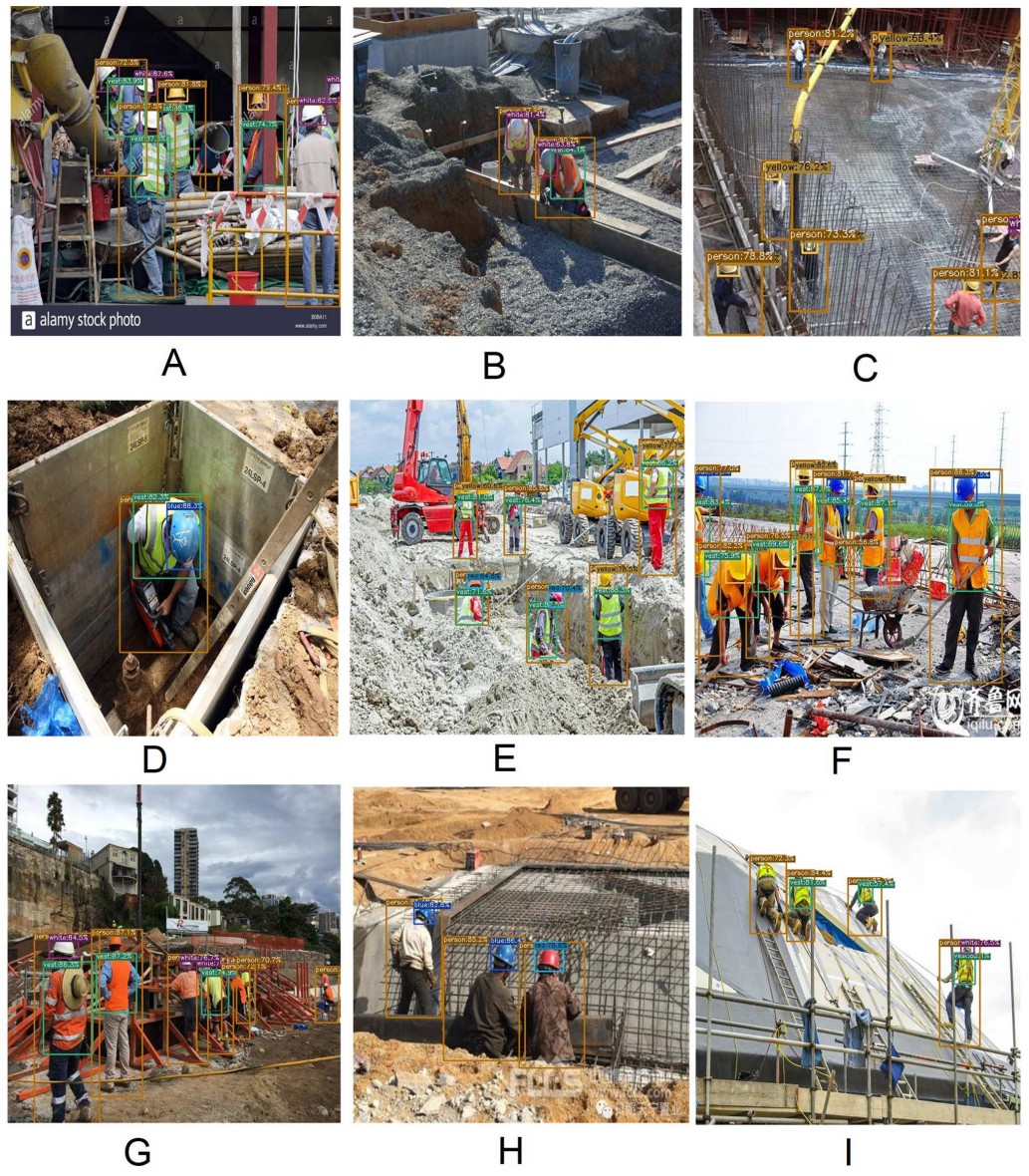

**Figure 12** **Satisfactory result of YOLOX-m architecture.** (A) Objects are in dense condition. (B, C, H) Objects are small. (D, E, F, I) Objects are in natural working conditions.

Most similar work like this study (*Wang et al., 2021b*) detect only six classes and the authors found that YOLOv5x yields the highest mAP of 86.55% for their CHV dataset. Though this study increases the dataset by adding extra two classes for detection, the YOLOX-m performs approximately 3.29% better than *Wang et al. (2021b)*. Training using low-light conditions, fog graduation, and rainwater image is our further investigation. However, this study linearly reduces the brightness of every pixel to make the low-light conditioned images. Moreover, different angled lighting condition and work with night mode images is our future investigation. In future work, safety boots and gloves can be included for further

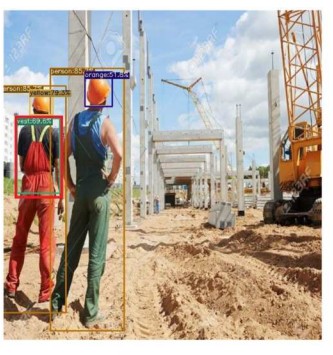 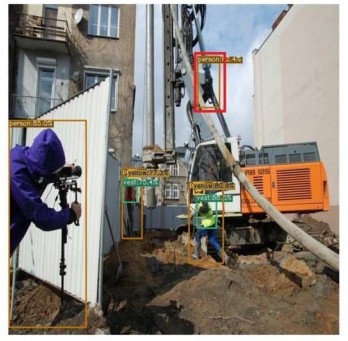 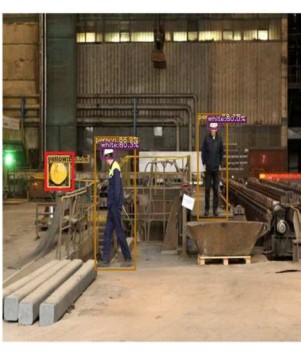

A                           B                           C

**Figure 13    Incorrect detection of YOLOX-m architecture.** (A) Something is detected as a vest which is not a vest. (B) An object is detected as a person which is not a person object. (C) A yellow color part of a machine is detected as a yellow hardhat.

**Table 5    Performance of YOLOX-m in the low-light, hazy and rainy effect image.**

| Category | Person | Head | Vest | Red | Yellow | Blue | White | Glass | mAP |
|---|---|---|---|---|---|---|---|---|---|
| Low-light | 90.01 | 89.77 | 87.12 | 85.06 | 87.66 | 86.82 | 93.47 | 70.79 | 86.33 |
| Hazy | 90.55 | 88.97 | 88.27 | 85.20 | 88.12 | 87.09 | 92.70 | 82.74 | 87.95 |
| Rainy | 87.96 | 80.16 | 80.86 | 75.63 | 78.36 | 78.20 | 73.35 | 69.38 | 77.98 |

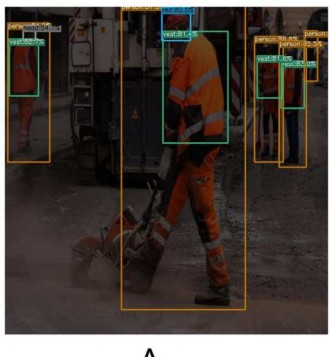 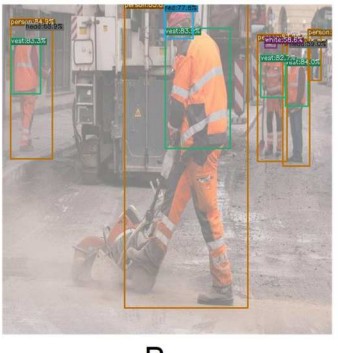 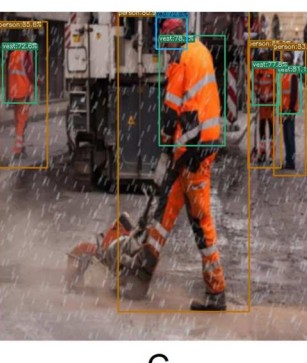

A                           B                           C

**Figure 14    Several qualitative measures of low-light, hazy and rainy effect images on YOLOX-m.**

exploration. Several person objects remain under-detected when they are more occluded. Safety glasses are transparent. For this reason, when they are small and blurry remain under-detected. Moreover, yellow parts of machines are detected as the yellow hard hat. The proposed model performs better than any previous construction site model. Hence, it can detect PPE on construction sites, which reduces time and cost in the industrial area.

**Table 6** Comparison between YOLOX-m and YOLOv5-x (*Wang et al., 2021b*) based on the same dataset and this study.

| Dataset | CHV | | CHVG* | |
|---|---|---|---|---|
| Method | YOLOv5-x | YOLOX-m* | YOLOv5-x | YOLOX-m* |
| person | 83.77 | 95.59 | 88.4 | 94.72 |
| head | × | × | 85.5 | 89.85 |
| vest | 81.47 | 85.94 | 91.5 | 90.28 |
| blue | 80.76 | 91.98 | 88.7 | 88.08 |
| red | 91.91 | 83.26 | 85.5 | 85.54 |
| yellow | 91.41 | 87.95 | 92.6 | 92.06 |
| white | 89.96 | 95.65 | 79.9 | 93.17 |
| glass | × | × | 77.8 | 85.00 |
| mAP | 86.55 | **90.06** | 86.24 | **89.84** |

**Notes.**
An asterisk (*) indicated values from this study. × Not available.

**Table 7** Blurring face test.

| Dataset | CHV | CHVG* |
|---|---|---|
| Model | YOLOv5-x | YOLOX-m* |
| Person | 82.78 | 91.83 |
| head | × | 88.97 |
| Vest | 82.46 | 88.27 |
| Red | 74.59 | 85.20 |
| Yellow | 85.17 | 86.12 |
| Blue | 73.78 | 87.09 |
| White | 78.46 | 93.70 |
| mAP | 79.55 | **88.73** |

**Notes.**
An asterisk (*) indicates values from this study. × Not available.

## CONCLUSION

The use of safety gear can protect them from unwanted accidents on a construction site. According to the statistics, every year tens and thousands of workers are tremendously injured at construction sites, creating lifelong difficulties. However, it is immensely important to make sure that the workers wear PPE by monitoring themselves. In this regard, an accurate and rapid system is needed to detect whether the workers are using PPE at the construction site. We created a new dataset named CHVG by extending the CHV (*Wang et al., 2021b*) dataset. An anchor-free training mechanism-based CV architecture is used in this study named YOLOX-m which yields the highest mAP 89.84% than another state-of-the-art method. The authors believe the proposed model can be implemented in real-time situations and industrial applications.

### Funding

This work was supported by the Khulna University of Engineering & Technology (KUET). The funders had no role in study design, data collection and analysis, decision to publish, or preparation of the manuscript.

### Grant Disclosures

The following grant information was disclosed by the authors:
The Khulna University of Engineering & Technology (KUET).

### Competing Interests

The authors declare there are no competing interests.

### Author Contributions

- Md. Ferdous conceived and designed the experiments, performed the experiments, analyzed the data, performed the computation work, prepared figures and/or tables, and approved the final draft.
- Sk. Md. Masudul Ahsan analyzed the data, prepared figures and/or tables, authored or reviewed drafts of the article, and approved the final draft.

### Data Availability

The code is available at GitHub: https://github.com/Md-Ferdous/YOLOX.

The data is available at figshare: Ferdous, Md.; Ahsan, Sk. Md. Masudul (2022): CHVG Dataset. figshare. Dataset. https://doi.org/10.6084/m9.figshare.19625166.v1.

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
