# Peer review of "PPE detector: a YOLO-based architecture to detect personal protective equipment (PPE) for construction sites"

_PeerJ Computer Science, doi:10.7717/peerj-cs.999_

## Round 0.1 · original submission · Major Revisions

Revise as per reviewers’ comments

Reviewer 1 ·

Basic reporting

The paper is, in general, written well, used formalism is explained, and the problem is stated clearly. Also, the application seems to be interesting. Unfortunately, there are three points that need to be addressed.

a) Abstract: the sentence "For the detection algorithm, this study has used the You Only Look Once (YOLO) family's anchor-free architecture, YOLOX, which yields better performance than the other object detection models within a satisfactory time interval." is not valid. The study does not bring a comparison with other SOTA detectors. The only comparison is made with YOLOv5-x. Based on general knowledge and benchmarks such as https://paperswithcode.com/sota/object-detection-on-coco benchmark, scaled YOLOv4 should yield better performance than YOLOx. The currently best detector is Swin, which still performs with a reasonable time.

b) Section 4: there is no motivation to implement your own haze/rain/low-light augmentations when all of them and many more are available through Albumentations? See https://albumentations-demo.herokuapp.com/ Based on it, Section 4 is useless and can be removed.

c) Section 5: it describes widely used evaluation metrics that are a gold standard in object detection and are well known by researchers. Therefore, there is no need to explain it in detail. It is enough to state you use IOU and mAP; thus, the section can be removed.

Experimental design

no comment

Validity of the findings

Here, it is necessary to state that two major issues were found. It is needed to address them carefully, mainly the second issue.

a) Repository:
- There is no readme in the repository regarding the described functionality. There is only a general readme forked from the original YOLOx repository. It must be described how to run the training script (which one is it) to replicate your results.
- I did not find training data or link to them.
- The model_test.ipynb functionality in the repository is nothing but a long error report. In the end, there is a test on seven images only without their visualization or mAP evaluation.
In summary, the correctness of the scheme cannot be confirmed.

b) Data: there is a question about the correctness of the labels. I examined several images from 'test-n.zip' and observed that:
- gettyimages-88655418-612x612_jpg.rf.6f0a5bd6a8ac6cf7a77d8a24b814e8df there are two person, two glasses, two helmets, and two vests. Label includes only one glass and one person.
- ppe_0579_jpg.rf.17a45729452bdb75971919d2630f0e21.jpg one orange helmet and one head are missing in the label
- ppe_0994_jpg.rf.98234bc61dda7a1fb4e54e7e02ade2a0.jpg the label for the bottom-right person is missing
- vitolda-klein-lAqSzwr5eQc-unsplash_jpg.rf.774701a69123aa712e43b4fc4deb1ed0.jpg the label misses orange helmet
- 000221_jpg.rf.8dff87700f9373a3eff79e1f9f55f273.jpg label person is missing for all people. That holds for all (8) images from this 'subseries.'
- gettyimages-83455052-612x612_jpg.rf.ea5cd5e8c3044737116b908a6139fd6b.jpg the label misses two helmets
...and many more.

Based on this fact, the results published in the papers are not trustworthy because they are computed for a highly noisy and inconsistent dataset. The dataset must be fixed, and the whole experiment section must be recalculated.

Additional comments

There are three minor comments:

a) Is 'Md. Ferdous' full name?

b) The sentence "In construction sites more than 71% injury appears than 32 in all other industries." needs to be rephrased.

c) Formula 4: there is written 'Where, G and P are the prediction and ground truth bounding boxes respectively'. It is formally correct, but it is better to mark P as prediction and G as ground truth and not contrariwise.

Reviewer 2 ·

Basic reporting

This review paper demonstrates a YOLO-based architecture to detect personal protective equipment for construction sites. The structure of the article is organized well however, some changes need to be done to improve it.

Some suggestions for further improvement:

The Introduction section is well-written, and I propose to schematize all the discussion in several paragraphs (to facilitate the reading): (1) motivations, (2) the overall approach, (3) main contributions.
However, the main challenges to the field and the needs and benefits of this study are missed in the introduction.

A general discussion of the limitations and expectations of the proposed model should be inserted.

Experimental design

Adding some effects such as haze or low light on images, will not change the features of the image significantly. I suggest creating a night-time dataset and adding it to the existing one.

The authors mentioned that the effected images are used only for testing. What about the original images of the effected images? If the same original images are used for training, then testing with the same images with just a small effect is not a good idea because in that case, the model will just memorize the images.

Please specify what is the main differences between CHV and CHVG datasets. Is it the same datasets just the number of classes are increased or the HAZE, RAIN, and LOW-LIGHT images are included as well?

Has exactly the same YOLOX model been used in this article or have the authors made any changes to it? This needs to be explained in the paper clearly.

Line 120: there are two “also” which are not necessary and can be removed.

Line 146: Please explain why DarkNet53 has been used as pre-trained weights in this model.

Line 186 and 188: Authors need to determine why they set these parameters.

Line 194: The image size needed to be explained in the dataset section.

Line 259: Please determine why the NMS value is set to 0.65

Validity of the findings

The evaluation metrics section is explained very well, however, the discussion section is too short and needs to be expanded.

It would be good if authors upload their code on GitHub and add the link in the paper.

---

## Round 0.2 · accepted · Accept

This article is accepted as per the reviewer comments

Reviewer 1 ·

Basic reporting

I appreciate that the authors carefully processed all my comments and improved the paper significantly. It is clear, easily readable, and correct.

Experimental design

I have no critical comments in this section.

Validity of the findings

The authors fixed the issue with the dataset labels. The findings are now supported by the data and I have no more critical comments.

Reviewer 2 ·

Basic reporting

No Comment

Experimental design

No Comment

Validity of the findings

No Comment